# Effect of Methionine on Gene Expression in *Komagataella phaffii* Cells

**DOI:** 10.3390/microorganisms11040877

**Published:** 2023-03-29

**Authors:** Tatiana Ianshina, Anton Sidorin, Kristina Petrova, Maria Shubert, Anastasiya Makeeva, Elena Sambuk, Anastasiya Govdi, Andrey Rumyantsev, Marina Padkina

**Affiliations:** 1Laboratory of Biochemical Genetics, Department of Genetics and Biotechnology, Saint Petersburg State University (SPBU), Saint Petersburg 199034, Russia; 2Institute of Chemistry, Saint Petersburg State University (SPBU), Petergof, Saint Petersburg 198504, Russia

**Keywords:** *Pichia pastoris*, *Komagataella phaffii*, X-33, methionine, amino acid metabolism, fatty acid metabolism, methanol metabolism, *AOX1* promoter

## Abstract

*Komagataella phaffii* yeast plays a prominent role in modern biotechnology as a recombinant protein producer. For efficient use of this yeast, it is essential to study the effects of different media components on its growth and gene expression. We investigated the effect of methionine on gene expression in *K. phaffii* cells using RNA-seq analysis. Several gene groups exhibited altered expression when *K. phaffii* cells were cultured in a medium with methanol and methionine, compared to a medium without this amino acid. Methionine primarily affects the expression of genes involved in its biosynthesis, fatty acid metabolism, and methanol utilization. The *AOX1* gene promoter, which is widely used for heterologous expression in *K. phaffii,* is downregulated in methionine-containing media. Despite great progress in the development of *K. phaffii* strain engineering techniques, a sensitive adjustment of cultivation conditions is required to achieve a high yield of the target product. The revealed effect of methionine on *K. phaffii* gene expression is important for optimizing media recipes and cultivation strategies aimed at maximizing the efficiency of recombinant product synthesis.

## 1. Introduction

The yeast *Komagataella phaffii* (previously known as *Pichia pastoris*) is one of the most efficient and robust systems for recombinant protein production. Powerful promoters, the high preference for respiratory growth, and the ability to rewire metabolic pathways for maximizing recombinant protein synthesis make *K. phaffii* a promising expression host [1,2,3]. *K. phaffii* performs all eukaryotic posttranslational modifications, and a weak tendency to protein hypermannosylation makes it a preferable producer in pharmacology to the model yeast *Saccharomyces cerevisiae* [4]. This feature goes hand in hand with successes in engineering its glycosylation pathways, and allows the production of proteins with modifications highly similar to mammalian cells [4,5]. Another beneficial feature is the ability of this yeast to efficiently synthesize and secrete proteins with a relatively easy subsequent purification procedure [6]. The availability of simple and robust high-cell density cultivation procedures and well-developed tools for strain engineering contribute to expanding the range of *K. phaffii* applications. Although many efficient molecular tools and techniques are now available to work on *K. phaffii*, achieving a high yield of the target product requires a sensitive adjustment of its cultivation conditions, which differ according to the target protein [7,8,9].

Thus, unique biological features, combined with long-term work on improving productivity, have made *K. phaffii* highly prominent for biotechnological and pharmaceutical applications [7]. To date, this yeast is applied for synthesizing thousands of recombinant proteins, both on a laboratory and industrial scale [9]. *K. phaffii* is used to produce recombinant subunit vaccines [10], various membrane proteins [11,12,13], and several nonprotein substances [14,15]. This practical importance stimulates the consideration of *K. phaffii* as a new model object for fundamental research, apart from being a production host [16].

Being methylotrophic, *K. phaffii* can grow on methanol as a sole carbon and energy source. The first reaction of the methanol utilization pathway (MUP) is the oxidation of methanol into formaldehyde by the enzyme alcohol oxidase, encoded by the *AOX1* and *AOX2* genes [17].

Part of formaldehyde is oxidized to CO_2_ in a series of reactions controlled by formaldehyde dehydrogenase, S-formylglutathione hydrolase, and formate dehydrogenase. This dissimilative branch of the MUP produces energy in the form of NADH, and prevents cell damage from the harmful effects of formaldehyde [18,19].

Another part of formaldehyde enters the assimilative branch of the MUP. Dihydroxyacetone synthase catalyzes the synthesis of dihydroxyacetone and glyceraldehyde-3-phosphate from formaldehyde and xylulose-5-phosphate. In further reactions, the enzymes dihydroxyacetone kinase, fructose-1,6-bisphosphate aldolase, and fructose-1,6-bisphosphatase result the formation of glyceraldehyde 3-phosphate, which is used for the generation of biomass and energy [18].

The product of the *AOX1* gene primarily has methanol-oxidizing activity in methanol-grown *K. phaffii* cells [17]. This gene expression depends on the type of carbon source in the medium [20]. For instance, glycerol or glucose both fully inhibit the *AOX1* promoter activity. On the contrary, *AOX1* gene expression is induced under conditions when methanol is the sole carbon source in the medium, and the level of *AOX1* mRNA reaches 5% of the total cell mRNA [17,18,19,20,21]. The strict regulation and high activity of the *AOX1* promoter determine its wide use for recombinant protein production [1]. This promoter is a part of plasmid vectors that are widely used for heterologous expression in *K. phaffii* (e.g., pPICZ series and pPIC9). Such practical importance drives molecular mechanism studies of the regulation of the *AOX1* gene and other methanol metabolism genes (methanol utilization, *MUT* genes) [22].

We have previously shown that the expression of *AOX1* and other *MUT* genes in *K. phaffii* cells is downregulated when certain amino acids are the sole medium nitrogen source [23,24]. This may be explained by the fact that this yeast can use several amino acids as the sole carbon, nitrogen, and energy sources [25,26]. Transcriptome analysis revealed differential expression of about 18.9% of all protein-coding genes when *K. phaffii* was grown on proline-containing media. It was shown that *K. phaffii* catabolizes proline, even when other nitrogen, carbon, and energy sources are available. As a result, the utilization genes of other sources, in particular methanol, are repressed [27]. Here, we continue these studies and investigate the effects of methionine as another amino acid.

Methionine and its derivatives implement various cell functions, such as DNA methylation, lipid homeostasis, and polyamine synthesis [28,29]. In yeast and mammalian cells, S-adenosylmethionine (a methionine derivative) has been shown to directly activate the target of rapamycin (TOR) complex I [30]. Methionine is of particular interest for its effect on longevity, given a wide range of studies showing that methionine restriction prolongs the lifespan in most model organisms [31].

Methionine plays an important role in the oxidative stress response, as it scavenges free radicals and protects the cell from reactive oxygen species (ROS) through participating in the activation of antioxidant enzymes [32]. Methionine derivatives are also a part of the biosynthesis of glutathione, the functions of which include counteracting oxidative stress [33].

Methionine stands out from other amino acids, since it acts as a translation initiation codon. Methionyl-tRNAi is required for translation initiation complex assembly [34]. In yeast cells, methionine activates the expression of genes involved in translation and ribosomal biogenesis [35].

Methionine metabolism has been studied in detail in *Saccharomyces cerevisiae* budding yeast. This microorganism can absorb methionine by specific and nonspecific permeases, and use the amino acid as the sole carbon source [36]. Additionally, *S. cerevisiae* can synthesize methionine de novo. The biosynthesis of this sulfur-containing amino acid involves three parts: construction of the methionine carbon chain, production of the sulfur anion, and incorporation of the sulfur anion into the carbon skeleton [37].

Aspartate is the precursor of the methionine carbon chain. Aspartokinase and aspartate-βsemialdehyde dehydrogenase catalyze the production of homoserine from aspartate. Next, homoserine O-acetyltransferase catalyzes the esterification reaction of homoserine and acetyl-CoA to produce O-acetylhomoserine [38].

Further addition of the sulfur atom is possible only in the form of sulfide, so sulfate and sulfite from the external environment are first reduced in the highly conservative pathway [39]. Once external sulfate enters the cell, it is metabolized into active adenosyl sulfate by ATP sulfurylase, and further converted to phosphoadenylsulfate (APS) by APS kinase. The enzymes PAPS reductase and sulfite reductase reduce APS to sulfite and sulfide, respectively [40].

Finally, homocysteine synthase integrates sulfide into O-acetylhomoserine to produce homocysteine. In yeast, this is the only reaction to incorporate sulfur into the carbon chain [37]. Homocysteine is a precursor of both methionine and cysteine, and the latter can be converted back into homocysteine via two transsulfuration pathways. To form methionine, the enzyme homocysteine methyltransferase removes the methyl group from 5-methyl-tetrahydrofolate (a one-carbon group compound) to methylate homocysteine [37].

One of the most essential methionine derivatives is S-adenosylmethionine (SAM), a major and vital cellular methylating agent. SAM synthetase catalyzes the SAM formation from ATP and methionine [41]. The methyl group transfer from SAM to the acceptor leads to S-adenosylhomocysteine (SAH) formation, which is further hydrolyzed to homocysteine. The resulting cycle, involving homocysteine, methionine, SAH, and SAM, is called the methyl cycle [40]. In addition, after SAM decarboxylation, the aminopropyl group can be involved in polyamine synthesis. As a result, methylthioadenosine (MTA) is formed, which can be converted into methionine again, closing the methylthioadenosine cycle [37]. In addition, SAM serves as an amino group donor in the synthesis of the biotin precursor pelargonium [42], and participates in the synthesis of modified nucleotides in tRNA and rRNA [43].

In *S. cerevisiae* yeast, methionine in the medium acts as a strong growth signal, triggering anabolism by activating the pentose phosphate pathway, amino acids, and nucleotide biosynthesis [35,44]. Thus, methionine and its derivatives play an important role in cellular metabolism. This research aims to study the complex effect of methionine on gene expression in the biotechnologically important yeast *K. phaffii*.

## 2. Materials and Methods

### 2.1. Media and Cultivation Conditions

Low salt LB medium was used for *Escherichia coli* manipulations. One L of low salt LB contained (here and further *w*/*v*): 1% tryptone, 0.5% yeast extract, 0.5% NaCl, 2.4% agar, and 20 mg zeocin for plasmid selection. Bacterial cells were grown at 37 °C.

YPD medium was used for *K. phaffii* manipulations. One L of YPD contained: 2% glucose, 2% peptone, 1% yeast extract, and 2.4% agar. YPDS media was used for zeocin selection. One L of YPDS contained: 2% glucose, 2% peptone, 1% yeast extract, 18.2% sorbitol (1 M), 2.4% agar, and 200 mg zeocin.

Modifications of standard buffered complex media were used for *K. phaffii* cultivation, transcriptome analysis, and quantitative enzymatic assays. One L of each media type contained: 100 mM potassium phosphate buffer (pH 6.0), 4 × 10^5^% biotin, and 1.34% Yeast Nitrogen Base without amino acids (#Y0626-1 KG, Sigma-Aldrich, St. Louis., MO, USA). BMG medium contained 1% glycerol. BMM medium contained 0.5% methanol. BMM and Met medium contained 0.5% methanol and 0.46% methionine. Yeast cells were grown at 30 °C.

### 2.2. Yeast and Bacterial Strains

The bacterial strain Escherichia coli DH5α F^−^ φ80*lac*ZΔM15 Δ(*lac*ZYA-*arg*F)U169 *rec*A1 *end*A1 *hsd*R17(r_K_^−^, m_K_^+^) *pho*A *sup*E44 λ^−^*thi*-1 *gyr*A96 *rel*A1 (Thermo Fisher Scientific, Waltham, MA, USA) was used for the construction of the pPICZ-PHO1 plasmid.

*K. phaffii* X-33 strain (Thermo Fisher Scientific, Waltham, MA, USA) was used for growth studies and transcriptome analysis.

*K. phaffii* PAP1-X33 strain was generated in this study. A DNA fragment corresponding to the coding sequence of *K. phaffii PHO1* acid phosphatase (ACP) gene (PAS_chr2-1_0103) was amplified by PCR using the Thersus Polymerase Kit (Evrogen, Moscow, Russia). Twenty picoM of primers PHO1-BstBI-F (5′-ATTACATTCGAAACGATGTTTTCTCCTATTCTAAGTCTGG-3′) and PHO1-AgeI-R (5′-ATTACTACCGGTTTATGACAAGTCATCCCAGAAG-3′) was added per reaction mix. The template was 500 ng of genomic DNA of X-33 strain. The reaction parameters were as follows: initial denaturation at 95 °C for 3 min, then 30 cycles 95 °C for 30 s, 53 °C for 30 s, 72 °C for 2 min. The resulting fragment was cloned into the pPICZαC plasmid using AgeI and BstBI restriction sites. The structure of the obtained pPICZ-PHO1 plasmid was confirmed by digestion with restriction enzymes and PCR analysis (Appendix A in Appendix A). The map of the pPICZ-PHO1 plasmid is depicted in Appendix A in Appendix A. The plasmid was linearized with the restriction enzyme SacI and transformed into *K. phaffii* X-33 strain. The resulting PAP1-X33 strain carries a *PHO1* reporter gene under the control of the *AOX1* gene promoter.

### 2.3. Molecular Methods

Yeast genomic DNA was extracted using LumiPure from an AnySample Kit (Lumiprobe, Moscow, Russia). Plasmids were isolated using the Plasmid Miniprep kit (Evrogen, Moscow, Russia). DNA was purified from agarose gels using a Cleanup Standard kit (Evrogen, Moscow, Russia).

DNA hydrolysis by restriction endonucleases, vector dephosphorylation, and DNA ligation were performed using the buffers and conditions recommended by the enzyme manufacturer (SibEnzyme, Moscow, Russia). PCR was performed using the Thersus Polymerase Kit (Evrogen, Moscow, Russia).

Yeast and bacterial cell transformations were performed using electroporation in accordance with the protocols [45,46].

### 2.4. Enzymatic Assays

A quantitative assay of acid phosphatase (ACP) activity was performed according to the previous research [47]. In brief, 100 μL of the cell suspension was taken and added into 800 μL of 0.1 M Na-citrate buffer, pH 4.6. Next, 100 μL of 0.15 M p-nitrophenyl phosphate substrate was added. The reaction mixture was stirred and incubated for 20 min at 30 °C in a water bath. The reaction was stopped by adding 500 μL of 1 M NaOH. The light absorption of the solutions was measured at a wavelength of 410 nm using a spectrophotometer. The specific activity of ACP was determined as the ratio of the absorption of light by the reaction mixture at a wavelength of 410 nm to the optical density of the initial cell suspension at a wavelength of 600 nm.

### 2.5. RNA-Sequencing

The RNeasy Mini Kit (Quiagen, Germantown, ML, USA) was used for the total RNA isolation from yeast cells. DNA molecules were removed from the sample by DNase (Thermo Fisher Scientific, Waltham, MO, USA) treatment. RNA purification was performed using the CleanRNA Standard Kit (Evrogen, Moscow, Russia). The quality of the isolated RNA molecules was assessed using agarose gel electrophoresis.

RNA-Seq library preparation was performed using the QuantSeq 3′ mRNA-Seq Library Prep Kit FWD (Lexogen, Vienna, Austria). The quality of the resulting libraries was tested with the Fragment Analyzer. The pool of libraries was sequenced on an Illumina MiSeq (single read, 300 bp) at Evrogen, Moscow, Russia.

### 2.6. Bioinformatic Analysis

The quality of the raw and trimmed reads was estimated using the FastQC program [48]. Illumina indices were removed and reads were filtered using the Trimmomatic program [49]. The quality-tested reads were aligned against the reference genome of *K. phaffii* yeast (ASM2700v1). The reference genome and annotations were retrieved from the NCBI database. Reads per genome were aligned using the hisat-2 program with standard parameters [50]. After alignment, reads were sorted and indexed using the samtools program [51]. The number of gene-aligned reads was calculated using the featureCounts program [52]. The number of resulting reads for each sample analyzed by RNA-sequencing is presented in Appendix A in the Appendix A.

Differential gene expression analysis was performed according to the standard protocol in the DESeq2 package (version 1.24.0) in the R Studio program using R language (version 3.6.3) [53,54]. Further analysis included differentially expressed genes with corrected *p*-value less than 0.05 and log2FoldChange greater than 0.5 modulo. The results of the DESeq2 analysis are presented in Appendix A in the Appendix A.

### 2.7. Gene Nomenclature

Names of *K. phaffii* genes with known annotation genes (*AOX1*, *AOX2*, etc.) were taken from other studies, especially [55]. For *K. phaffii* genes that were analyzed in this study as orthologs of the *S. cerevisiae* genes, “*Kp*” index was added to the name to distinguish them from *S. cerevisiae* ones.

## 3. Results

### 3.1. Addition of Methionine Does Not Enhance Growth of K. phaffii X-33 Strain

It was shown that some amino acids (e.g., proline and glutamate) stimulate *K. phaffii* growth in media containing methanol [23]. This is because the utilization of these amino acids is enough to provide *K. phaffii* cells with carbon, nitrogen, and energy [25,26]. As for methionine, it was shown that *K. phaffii* is not able to grow in media with this amino acid as the sole carbon, nitrogen, and energy source [56]. However, it was revealed that external methionine triggers anabolism and proliferation in *S. cerevisiae* [35]. Thus, we investigated if methionine affects growth of *K. phaffii* in media with glycerol and methanol as carbon and energy sources.

The starting culture of *K. phaffii* wild-type X-33 strain was grown overnight in 20 mL of liquid YPD media. Volumes of 100 mL of BMG, BMG and Met, BMM, BMM and Met, and BM and Met media were inoculated with 0.1% (100 μL) of starting culture. The optical density of yeast cultures was measured with a spectrophotometer at a wavelength of 600 nm (Figure 1).

*K. phaffii* X-33 strain did not grow in BM and Met media with methionine as a sole source of carbon and energy, which supports previous data [56]. In glycerol-containing BMG media, addition of methionine did not significantly influence the growth of *K. phaffii*. However, in methanol-containing BMM media, addition of methionine resulted in a slight decrease in cell density. Thereby, we also investigated how methionine affects gene expression in *K. phaffii* cells.

### 3.2. Methionine Influences the Expression of Genes Involved in Its Synthesis and Degradation in K. phaffii Cells

For the experiment, *K. phaffii* cells were grown according to a two-stage cultivation scheme, where at the first stage the cells grew biomass in a medium with glycerol, and then were transferred to a medium with methanol, which leads to the activation of MUT genes. Thus, the *K. phaffii* X-33 strain was grown in 100 mL of BMG medium with glycerol. After 40 h of growth, the cells were transferred to 100 mL of the medium with methanol and methionine, or without methionine (BMM and Met, or BMM). After 15 h of growth in the methanol-containing medium, total RNA was isolated from the cells and further used to prepare libraries for RNA sequencing.

Differential gene expression analysis revealed 142 differentially expressed genes (DEGs), which is 3% of the total number of transcribed genes detected (4687 protein-coding genes) (Appendix A in the Appendix A). Among them, 75 genes increased and 67 genes decreased expression in the methionine-containing medium compared with gene expression in the medium without methionine, respectively.

One of the main groups of DEGs comprises genes involved in methionine biosynthesis. Gene homology and metabolic flux profiling studies revealed that amino acids are synthesized with high similarity in *K. phaffii* and *S. cerevisiae* [57,58]. Therefore, we used *S. cerevisiae* data to match these DEGs to specific parts of the methionine metabolism pathway (Figure 2).

Methionine biosynthesis in *S. cerevisiae* comprises three main parts: construction of the methionine carbon chain, production of the sulfur anion, and incorporation of the sulfur anion into the carbon skeleton [37].

Our data demonstrate the repression of the sulfur assimilation branch genes in the methionine presence. The expression level of adenylylsulfate kinase (*KpMET14*), sulfite reductase β-subunit (*KpMET5*), and sulfite reductase α-subunit (*KpMET10*) was significantly decreased in methionine-containing media. On the other hand, expression of genes involved in O-acetylhomoserine biosynthesis did not change significantly in response to methionine.

The expression of the methionine and cysteine synthase gene (*KpMET25*), which connects the sulfur assimilation branch and O-acetylhomoserine, was reduced in media with methionine. The methionine synthase gene *KpMET6* did not change expression in methionine-containing media.

The expression level was significantly reduced for the folylpolyglutamate synthase gene *KpMET7*. This gene encodes an enzyme that controls the synthesis of tetrahydrofolate in the folate cycle. A derivative of this metabolite is involved in methionine synthesis. The cysteine biosynthesis gene encoding cystathionine gamma-lyase *KpCYS3* showed reduced expression in response to methionine in the medium.

### 3.3. Methionine Influences the Expression of Genes Involved in Fatty Acid Metabolism in K. phaffii Cells

Differential expression analysis revealed the suppression of de novo fatty acid (FA) synthesis by methionine in the culture media (Figure 3). In *S. cerevisiae,* the first and rate-limiting step in this biosynthesis process is a reaction catalyzed by acetyl-coenzyme A carboxylase [60]. This reaction requires a biotin cofactor that attaches to the apoprotein ACC1p [61]. Fatty acid chain synthesis continues by attaching acetyl-CoA and malonyl-CoA to the fatty acid synthase (FAS) complex domain, called acyl-carrier protein (Acp). Then acetoacetyl is formed in the decarboxylation reaction, and like the other acyl residues, it is bounded to Acp at every biosynthetic step. The FAS complex consists of two polyfunctional proteins, α and β, which are encoded by the *FAS2* and *FAS1* genes, respectively [62]. The FAS complex works throughout the fatty acids biosynthesis by adding two residues of malonyl-CoA to the growing chain.

According to our results, for *K. phaffii* cells grown in a methionine-containing medium, gene expression was reduced for the enzymes acetyl-CoA carboxylase (*KpACC1*) and fatty acid synthetase (*KpFAS1* and *KpFAS2*), which are crucial for fatty acid biosynthesis. The gene expression of the enzyme long-chain fatty acyl-CoA synthetase, with a preference for C12:0-C16:0 fatty acids (*KpFAA1*), which makes long-chain acyl-CoA, including palmitoyl-CoA, was also reduced.

On the contrary, the expression of the elongase enzyme gene (*KpELO*) of the elongation pathway (C20 residues and above) of FAs was significantly increased. This pathway occurs in the endoplasmic reticulum, where FAs up to 26 residues form. The reactions carried out by elongase are similar to those occurring in de novo biosynthesis, and require malonyl-CoA. FA-acyl-CoA to Acp bonding is not required for elongation pathway reactions [61].

Expression of the gene for the fatty acid β-oxidation enzyme fatty-acyl coenzyme A oxidase (*KpPOX1*) was also increased. In yeast, β-oxidation is performed only in the peroxisomes [64]. This process produces acetyl-CoA, which can further enter the glyoxylate cycle or pass into the mitochondria. However, the expression of the peroxisomal membrane protein required for the medium-chain fatty acid oxidation gene (*PEX11*), which is involved in β-oxidation and peroxisome proliferation, was reduced.

One of the fatty acid derivatives are phospholipids, components of cell membranes. They are formed from glycerol-3-phosphate backbone and fatty acid residues [65]. Our results show decreased expression of the *KpOPI3* gene. In *S. cerevisiae*, the *OPI3* gene product synthesizes phosphatidylcholine (PC) from phosphatidylethanolamine (PE) in reactions coupled with the consumption of SAM [66].

### 3.4. Methionine Influences the Expression of Genes Involved in Methanol Utilization in K. phaffii Cells

The activity of some methanol utilization (MUT) genes was decreased when *K. phaffii* grew in methionine-containing media (Figure 4).

It was shown that in a methionine-containing medium, the expression of the alcohol oxidase *AOX2* gene is decreased. Repression of the *AOX1* gene, which is responsible for 90% of alcohol oxidase, may be considered if a less strict cutoff for DEG is applied (0.05 < p adj < 0.1). Genes involved in first reactions of the dissimilative branch (*FLD1* and *FGH1*) do not change their expression. The formate dehydrogenase gene *FDH1* is repressed by methionine.

Genes encoding the enzymes of the assimilative branch of methanol metabolism also demonstrate different expression. Dihydroxyacetone synthase genes *DAS1* and *DAS2* are repressed in response to methionine in the media. Among other main genes, only *TPI1* may be considered repressed by methionine if a less strict cutoff for DEG is applied (0.05 < p adj < 0.1). Dihydroxyacetone kinase (*DAK1*), fructose-1,6-bisphosphate aldolase (*FBA1* and *FBA2*), and fructose-1,6-bisphosphatase (*FBP1*) did not significantly change their mRNA levels in response to methionine.

The expression level was reduced for genes involved in cellular defense mechanisms against reactive oxygen species that are usually produced in methanol oxidation processes. While catalase and superoxide dismutase genes (*CAT1*, *SOD1*) did not significantly change their expression, peroxisomal glutathione peroxidase gene *PMP20* was repressed by methionine.

Thus, in *K. phaffii* cells, *MUT* genes demonstrate different regulation in response to methionine. Some of them are downregulated when this amino acid is added to the media, but others do not significantly change their expression levels. Interestingly, the expression of the KpMIG1 gene, which is implicated in the repression of MUT genes [67], was upregulated when methionine was added to the medium.

### 3.5. AOX1 Gene Promoter Is Repressed in Medium with Methionine

Since transcriptome analysis demonstrated borderline results for *AOX1* gene expression, a reporter system was used to precisely analyze its expression in dependence on methionine. A *K. phaffii* PAP1-X33 strain carrying a *PHO1* reporter gene under the *AOX1* gene promoter control was generated. The resulting strain synthesizes and secretes acid phosphatase in response to activation of the *AOX1* promoter by methanol (Appendix A in the Appendix A).

PAP1-X33 strain was grown in 100 mL BMG media with glycerol. After 40 h of growth, cells were transferred to 100 mL of media with methanol, and with or without methionine (BMM and Met, or BMM). After 24 h of methanol induction, ACP specific activity in the cultures was measured (Figure 5).

It was shown that the activity of the *AOX1* promoter is indeed decreased during methanol induction if methionine is present in the medium.

## 4. Discussion

*K. phaffii* yeast plays a prominent role in modern biotechnology as a workhorse for the production of recombinant proteins and various chemical substances. For efficient use of this yeast, it is important to study the effects of different media components on its growth and gene expression. The amino acid methionine is not only a block for protein synthesis, but is also crucial for a large number of cell processes. For *S. cerevisiae* yeast, the presence of methionine in the media can induce a hierarchically organized anabolic program enabling proliferation [35].

*K. phaffii* can metabolize several amino acids (e.g., proline) as a sole carbon and energy source. The presence of such amino acids in the media stimulates the growth of *K. phaffii* cells and strongly regulates its gene expression [27]. *K. phaffii* cells do not use methionine as a carbon and energy source [55]. Our results demonstrate that it also does not significantly influence the growth of *K. phaffii* X-33 strain in media containing glycerol. In media with methanol, its growth was slightly impaired when methionine was added at a concentration of 0.46%. This coincides with the results of gene expression analysis that demonstrate repression of some *MUT* genes if methionine is present in the media during methanol induction.

Our previous results demonstrate that the presence of proline in the media results in the repression of all main *MUT* and *PEX* genes in *K. phaffii* cells [27]. Thus, a signal system downregulates the expression of *MUT* and *PEX* genes to an optimal level when proline and methanol are concurrently metabolized by *K. phaffii* cells. In this case, these genes are regulated as a unit, and form a regulon. However, for methionine the picture is different—only some of the *MUT* genes are downregulated, while expression of others does not depend on the presence of methionine in the media.

Several proteins were previously found to be directly involved in *AOX1* and *MUT* gene regulation. Here, we demonstrate that expression of the *KpMIG2* gene (*PAS_chr1-4_0526*), which is involved in the repression of *AOX1* and other *MUT* genes [67], is significantly upregulated when *K. phaffii* is grown in methionine-containing media. Interestingly, the expression of this gene is also activated by proline [27]. It might be suggested that this gene represents an important node that intertwines the regulation of methanol utilization and other metabolic processes in *K. phaffii* cells.

The effect of methionine on fatty acid and lipid metabolism in different cells has been reported, but the mechanism of this relationship has not yet been identified [29,68]. Our results demonstrate the complex effect of excess methionine on fatty acid metabolism in *K. phaffii* cells. According to the differential expression analysis, fatty acid de novo biosynthesis was suppressed in a medium with methionine. However, the genes of the first enzymes of the elongation and fatty acid β-oxidation pathway were upregulated under the influence of methionine. These results suggest that methionine redirects fatty acid metabolism.

It has been previously suggested that the tight regulation of fatty acid metabolism may occur because of the restricted availability of acetyl-CoA [69]. However, our data show a statistically significant increase in gene expression of the acetyl-coenzyme A synthetase enzyme (*PAS_chr3_0403*), as well as suppression of gene expression of the acetyl-CoA carboxylase (*PAS_chr1-4_0249*) that produces malonyl-CoA from acetyl-CoA, which is repressed by acyl-CoAs, including malonyl-CoA and palmitoyl-CoA [70]. In addition, in our case, there was increased expression of the fatty acid β-oxidation enzyme gene (*PAS_chr1-4_0538*), which results in the acetyl-CoA synthesis. All of the above suggests that acetyl-CoA deficiency is unlikely to cause a strong decrease in the gene expression of the fatty acid de novo biosynthesis pathway. Overall, finding a direct link between methionine and fatty acid metabolism seems to be problematic, based on transcriptomic data alone.

In *S. cerevisiae* cells, transcription of methionine biosynthesis pathway genes is suppressed by the presence of large amounts of methionine or cysteine in the media [71]. Our results showed that for *K. phaffii,* methionine presence in the media results in the downregulation of genes involved in the sulfur assimilation pathway. We also observed a decrease in expression of the *KpMET25* gene, which is responsible for a binding reaction between two branches of methionine biosynthesis: sulfur assimilation, and construction of methionine carbon backbone. The genes of the latter branch did not significantly change in their expression levels. This may occur because aspartate and homoserine are indirect precursors for other amino acids, e.g., threonine and isoleucine [72].

Media recipes for *K. phaffii* cultivation often contain peptone and amino acid mixes, and methionine is an essential part of these ingredients. Our results revealed the methionine effect on gene expression in *K. phaffii* cells, which is important for further optimization of this yeast cultivation. For instance, *K. phaffii* MutS strains have previously been shown to be prone to inappropriately incorporate O-methyl-L-homoserine instead of methionine residues when cells grew in culture media with methanol as the sole carbon source. Supplementation of the cultivation media with methionine is proposed to eliminate this negative effect [73]. Our results demonstrate that such supplementation should be carefully optimized to balance between reducing O-methyl-L-homoserine misincorporation and repressing activity of the *AOX1* gene promoter, which is often used for recombinant protein production.

*K. phaffii* yeast is now considered a prospective model object for fundamental research. This trend is stimulated by the practical importance *K. phaffii* and development of modern research techniques, such as methods of transcriptome, proteome, and metabolome analysis. *K. phaffii* possesses a sophisticated mechanism that regulates the expression of *AOX1* and other *MUT* genes. This mechanism is inextricably linked to regulation of different metabolic pathways and cellular processes. It allows *K. phaffii* to efficiently coordinate expression of *MUT* genes in response to various factors. Interestingly, most of the proteins involved in this regulation have orthologues in other yeast species that do not possess methanol utilization pathways. It may be proposed that, during the evolution of methylotrophic yeasts, preexisting regulatory elements rearranged and took under control the emerging unique pathway of methanol utilization. Thus, further investigation of *MUT* genes regulation in *K. phaffii,* in comparison to other yeast species, becomes important for understanding the evolution of regulatory systems. In this study, we demonstrate novel aspects of such regulation.

## Figures and Tables

**Figure 1 microorganisms-11-00877-f001:**
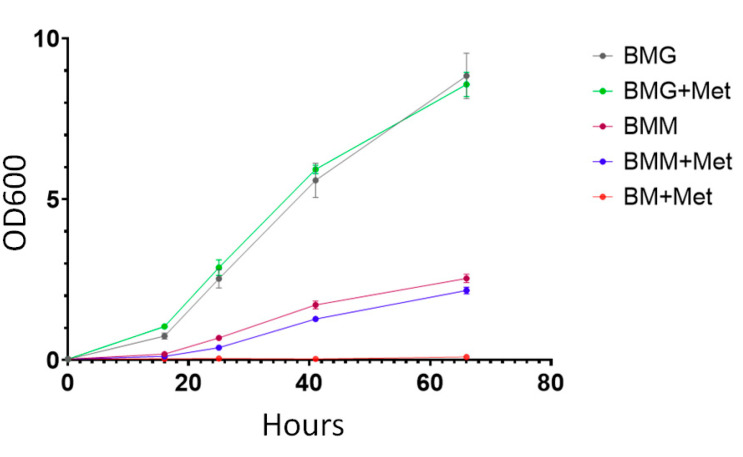
Growth curves of *K. phaffii* X-33 strain in different media. BMG medium contains 1% glycerol; BMG and Met—1% glycerol and 0.46% L-methionine; BMM—1% methanol; BMM and Met—1% glycerol and 0.46% L-methionine; BM and Met—only 0.46% L-methionine. Cells were grown in four separate cultures for each type of the medium. Each measurement was performed in two replicas for each culture. Mean OD600 values ± SEM are plotted. Raw data is presented in Appendix A in the Appendix A.

**Figure 2 microorganisms-11-00877-f002:**
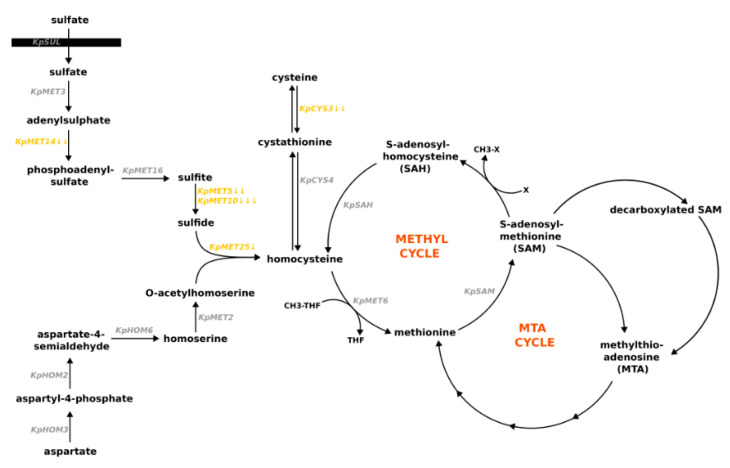
Scheme of proposed methionine metabolism pathways in *K. phaffii* cells (based on data for *S. cerevisiae* [37,59]). Genes are placed near the reactions catalyzed by the corresponding proteins. Genes that are repressed in media with methionine are marked in orange. Genes that do not change their expression are marked in grey. The number of arrows represents values of the log2FoldChange parameter: one arrow 0.5 < log2FoldChange < 1; two arrows 1 < log2FoldChange < 2; three arrows 2 < log2FoldChange.

**Figure 3 microorganisms-11-00877-f003:**
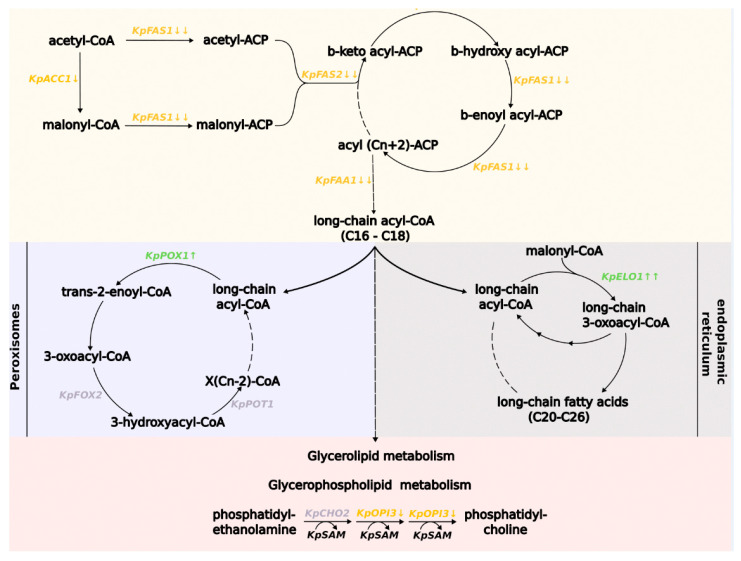
Scheme of the proposed fatty acid metabolism pathways in *K. phaffii* cells (based on data for *S. cerevisiae* [61,63]). Genes are placed near the reactions catalyzed by the corresponding proteins. Genes that are repressed in media with methionine are marked in orange. Genes that are activated in media with methionine are marked in green. Genes that do not change their expression are marked in grey. The number of arrows represents values of the log2FoldChange parameter: one arrow 0.5 < log2FoldChange < 1; two arrows 1 < log2FoldChange < 2; three arrows 2 < log2FoldChange.

**Figure 4 microorganisms-11-00877-f004:**
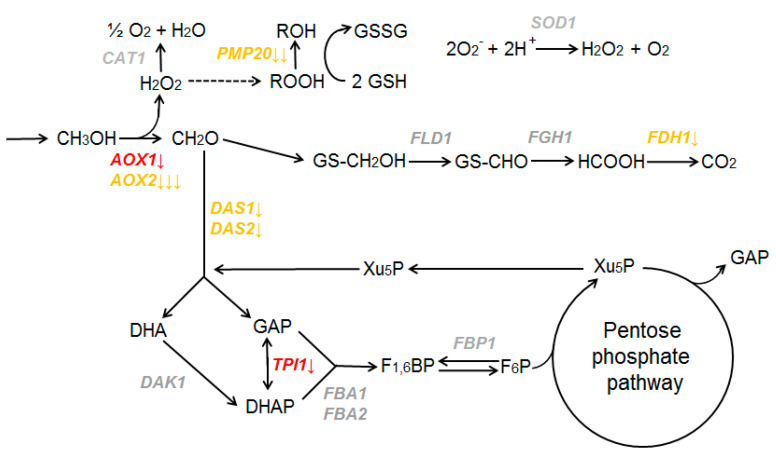
Scheme of the canonical methanol utilization pathway in *K. phaffii* cells (modified from [47]). Genes are placed near the reactions catalyzed by the corresponding proteins. Genes that are repressed in media with methionine are marked in orange (p adj < 0.05) and red (0.05 < p adj < 0.1). Genes that do not change their expression are marked in grey. The number of arrows represents values of the log2FoldChange parameter: one arrow 0.5 < log2FoldChange < 1; two arrows 1 < log2FoldChange < 2; three arrows 2 < log2FoldChange. Abbreviations of metabolites: DHA, dihydroxyacetone; DHAP, dihydroxyacetone phosphate; FRU_1,6_BP, fructose-1,6-bisphosphate; F_6_P, fructose-6-phosphate; GAP, glyceraldehyde-3-phosphate; GSH, glutathione; GSSG, oxidized glutathione self-dimer; Xu5P, xylulose 5-phosphate.

**Figure 5 microorganisms-11-00877-f005:**
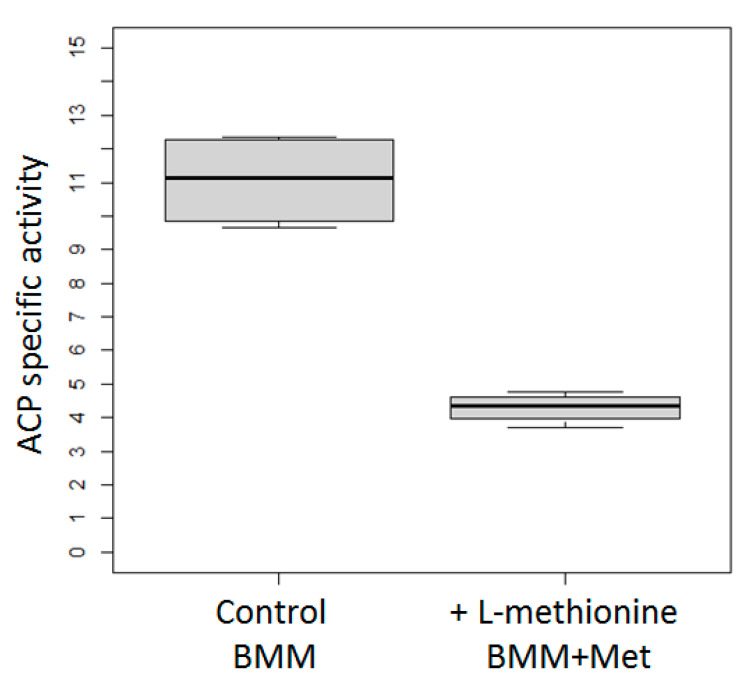
ACP specific activity in the cultures of *K. phaffii* PAP1-X33 strain after methanol induction in media with or without methionine (BMM and Met, or BMM). The mean from four separate experiments with two technical replicates is plotted. Raw data is presented in Appendix A in the Appendix A.

## Data Availability

The data presented in this study are available in Appendix A.

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
