# Peer review of "Effect of Methionine on Gene Expression in Komagataella phaffii Cells"

_microorganisms, 2023, doi:10.3390/microorganisms11040877_

Round 1

Reviewer 1 Report

In my opinion. The manuscripts suits into journal demands and may be publisehd without any further improvements

Author Response

To Whom It May Concern,

Thank you for thoroughly reviewing our paper. We appreciate your high assessment and time a lot.

Reviewer 2 Report

With the rapid progress of synthetic biology, optimizing cellular systems has become a crucial task. This manuscript presents potential applications obtained from optimizing the Komagataella phaffii yeast. The experimental design is clear, the process is rigorous, and the results are reliable. In general, the manuscript is of high quality.

The abstract should conclude by summarizing and describing the significance of the optimization of the medium formulation and culture strategy, which led to positive results.

In the introduction, it is essential to discuss the advantages and disadvantages of Komagataella phaffii yeast in comparison to other yeasts used for chassis cells, highlighting the unique advantages of using Komagataella phaffii yeast as a chassis cell, thereby increasing the importance of the research on this topic.

The discussion section should consider whether the induction effect of methanol and methionine on Komagataella phaffii yeast is similar to that of other engineering yeasts and if similar experiments have been performed in other yeasts.

In line 146, "rK-mK+" should be "rK-mK+," and "proAB+" should be "proAB+."

A reference or URL for the DESeq2 package (version 1.24.0) should be added in line 206.

All references should follow the formatting requirements and eliminate redundant forms.

Species names, including those in the literature, should be italicized, such as in lines 303, 521, 541, 561, 578, 581, 586, 606, 630, 636, 644, 647, 660, and 662. Additionally, the species name does not require abbreviations at the beginning of paragraphs and sentences, as in line 148.

Author Response

To Whom It May Concern,

Thank you for thoroughly reviewing our paper and appreciating our work. Please find our answers in the attachment. We look forward to hearing your opinion.

Reviewer 3 Report

In general, the article is characterized by its hypothesis testing. However, PCR conditions, including the number of cycles, concentration of reagents, and operating temperatures, were not included in the methodology. The article contains a good quantity and quality of bibliographic citations.

The manuscript is clear and relevant to the field of gene expression. It is well organized; however, most of the citations are not recent (within the last 5 years). Reference number 8 is missing the year of publication and the DOI code of the cited article. The experimental design used is appropriate for testing the hypothesis. The results of the manuscript are reproducible as specified in the methods section, except for the PCR conditions, which are not specified. The figures are generally appropriate, but it is suggested that Figure 5 be enlarged to better distinguish them. They are easy to interpret and understand and interpret the data appropriately and consistently throughout the manuscript. In lines 418 and 419, there is a redundancy in the sentence that needs to be revised.

Line 422 is missing the bibliographic reference that supports the given statement. The conclusions are consistent with the evidence and arguments presented. In addition, the conclusions are consistent and supported by the citations listed.

Author Response

(The authors gave the same response as above.)
